# Biological interaction of bioactive polymeric membranes in induced bone defects in rabbit tibias

Olicies da Cunha[1], Cássio Ricardo Auada Ferrigno[2]*, Solimar Dutra da Silveira[3], Bruno Gregnanin Pedron[4], Danielle Tiemi Komorizono[5], Flávia Cristina Rosin Prado[6], Walter Israel Rojas Cabrera[7], Luciana Corrêa[6]

1 Department of Veterinary Sciences, Federal University of Paraná, Curitiba, Palotina–PR, Brazil,
2 Department of Small Animal Clinical Sciences, University of Tennessee, Knoxville, TN, United States of America, 3 Graduate Program in Animal Science, Federal University of Paraná, Curitiba, Palotina–PR, Brazil,
4 Department of Anesthesiology, Teaching and Research Institute of Hospital Sírio-Libanês, São Paulo–SP, Brazil, 5 Private Sector, São Paulo–SP, Brazil, 6 Department of Stomatology, University of São Paulo, São Paulo–SP, Brazil, 7 University of São Paulo, São Paulo–SP, Brazil

ʘ These authors contributed equally to this work.
‡ These authors also contributed equally to this work.
* cferrign@utk.edu

**Data Availability Statement:** All relevant data are within the paper and its Supporting Information files.

## Abstract

The study aimed to evaluate bone repair using three osteoinductive polymers in bone defects created in rabbit tibias. Forty-eight adult rabbits were assessed at various time points: three, seven, fourteen, and thirty days. The groups included a control group (without biomaterial), M1 (Poly L Lactide co Polycaprolactone/Polyethylene Glycol), M2 (Poly L Lactide co Polycaprolactone/Polyethylene Glycol/β-Tricalcium Phosphate), and M3 (Poly L Lactide co Polycaprolactone/Polyethylene Glycol/nano hydroxyapatite). Histomorphometric analysis was conducted to evaluate new bone formation within and around the bone defect. At 14 ($p<0.05$) and 30 days ($p<0.05$), the callus area in the membrane groups, particularly in M3, was also significantly larger than in the control group, indicating the osteoinductive potential of these biomaterials. The callus consisted of both bone and cartilaginous matrix, suggesting a robust activation of endochondral ossification. The number of osteoclast was higher in the membrane groups, especially at 14 days in the M3 group, indicating increased bone remodeling activity. The membranes were not fully absorbed by 30 days, creating a space between the defect and the periosteum. In conclusion, all three membranes showed significant chondro and osteoinductive potential, with the membrane containing nano-hydroxyapatite demonstrating the most pronounced potential.

## Introduction

Bone repair follows a process similar to that of non-bone tissues repair, involving complex interactions between cells, growth factors, and extracellular matrix. Traditionally, it is divided into four stages: inflammation, granulation tissue formation, callus formation, and

**Funding:** The research was paartially funded by FUndacao de Amparo a Pesquisa do Estado de São Paulo. The funding pay for the animals used in the project, and the disposeble use in the project.

**Competing interests:** The authors have declared that no competing interests exist.

remodeling. However, in practice, these phases are not strictly delimited, and there is often overlap of events during the repair process [1–4].

MARTIN, GOOI, & SIMS (2009) [5] mentioned that in fracture repair, there is a balance between the anabolic (bone-forming) and catabolic (bone resorption) processes. The final stage of fracture repair involves bone remodeling. Initially, this process involves the conversion of irregular bone callus into lamellar bone tissue, which is essential for restoring the mechanical integrity of the bone [3,4].

Remodeling begins around the 14th day and can extend for months [3]. The primary cell involved in the resorption of mineralized bone is the osteoclast, which, when active, is polarized, with part of the membrane engaged in resorption [6]. The remodeled bone callus can have superior mechanical properties per unit area of bone compared to an incompletely remodeled bone callus, but a larger callus may provide similar mechanics properties [4].

Synthetic biodegradable membranes are designed to stimulate cells at the injury site and promote repair at the desired location [7,8]. They act as carriersfor bioactive substances, morphogenetic factors, anti-inflammatory agents, and antibiotics necessary to stimulate the biochemical repair processes and cellular signaling, accelerating the cellular response and creating favorable conditions for new tissue formation [9,10].

Currently, no published experimental studies evaluate the association of PLLA-co-PCL/PEG polymers without additives or with the addition of β-TCP or nano-HA. Therefore, the present study aims to assess microscopic aspects of bone repair in self-induced defects in rabbit tibias by implanting of polymer, polymer with β-TCP, and polymer with nano-HA. This is done with the intention of supporting their clinical use.

## Methods

### Ethical aspects

The research was conducted after approval by the 'Ethics Committee on Animal Use' of the Faculty of Veterinary Medicine and Animal Science of the University of São Paulo, with protocol number 1763/2009.

### Study design

Forty-eight New Zealand rabbits (*Oryctolagus cuniculus*), young adult males weighing between 3200 and 3500g, were studied. The animals underwent an adaptation period in a ventilated environment in individual cages, with free access to water and commercial feed. The environment was sanitized daily. The study was divided into experimental time points of three, seven, 14, and 30 days, with 12 patients randomly grouped at each experimental time point. The experimental times of three, seven, 14, and 30 days were designated to assess the potential inflammatory reaction induced by the biomaterials, particularly the foreign body reaction. The 14-day period was specifically chosen for morphometric analysis to evaluate the osteoinduction potential of the material, as this stage is critical in the bone regeneration process due to increased cell proliferation. The 30-day time frame was allocated for assessing the initial phase of bone remodeling induced by the membranes and to determine whether there was resorption of the biomaterial by osteoclasts or other types of phagocytic cell.

The tibias of the 12 rabbits in each period accounted for 24 units, with six for each treatment: control—only the hole without a membrane; membrane 1 (M1)—consisting of a polymer based on Poly L Lactide co Polycaprolactone / Polyethylene Glycol (PLLA-co-PCL/PEG); membrane 2 (M2)—polymer and β-Tricalcium Phosphate (PLLA-co-PCL/PEG/β-TCP); membrane 3 (M3)—polymer and nano-hydroxyapatite (PLLA-co-PCL/PEG/nano-HA).

The animals in the control group were subjected to the same anesthetic, pre-operative, transoperative, and post-surgical protocols. The only difference was the absence of any material in the created defects.

## Surgical procedure and sample collection

The animals were premedicated intramuscularly with ketamine (30 mg/kg) and midazolam (1 mg/kg), followed by mask induction with isoflurane. All animals underwent endotracheal intubation, and anesthesia was maintained with isoflurane in oxygen in a non-rebreathing circuit to achieve a surgical plane of anesthesia, characterized by muscle relaxation, ventral ocular rotation and absence of palpebral reflex. Sacrococcygeal epidural lidocaine 2% (0.3ml/kg) provided surgical analgesia.

Two 2.8 mm in diameter single-cortical holes were created in the proximal region of the medial aspect of the tibia, located 15 mm from the femoral-tibia-patellar joint, with a 10 mm spacing between them. Subsequently, a membrane was gently placed over the surgical hole. In 24 tibias, no membranes were placed, and these were part of the control group. The periosteum was repositioned over the membrane and sutured with absorbable 5–0 polyglycolic acid sutures, followed by suturing of the other tissue layers (Fig 1).

For each experimental time point, 12 rabbits or 24 tibias were used, resulting in six control specimens, six M1 specimens, six M2 specimens, and six M3 specimens.

At the end of the surgical procedure, all animals received ketoprofen (3mg/kg) and morphine (2mg/kg) subcutaneously for 5 and 3 days, respectively.

After the observation periods, the animals were euthanized, and the tibias were prepared to histological analyses. At each evaluation time point, the animals were sedated with intramuscular ketamine (10 mg/kg) and midazolam (1 mg/kg). Anesthesia induction was performed with sodium thiopental (50 mg/kg) intravenously, followed by euthanasia with a lethal dose of potassium chloride using the same route.

## Histomorphology analysis

After decalcification, the 96 tibias were cut lengthwise to divide the holes in half. This procedure created two sections (A and B) for each specimen and increased the number of samples. Additionally, it allowed for macroscopic evaluation (Fig 2). By dividing each of the 96 experimental units, 192 sections were obtained and evaluated macroscopically.

Eight slides were prepared for morphometric analysis under light optical microscopy, with four from each section (A and B). The sections, 5 μm in thickness, were made using a manual tabletop microtome with the assistance of a blade. Six slides were stained with H&E (Hematoxylin and Eosin), one with TRAP (Tartrate-Resistant Acid Phosphatase) staining, and one with Masson's Trichrome staining. Each specimen was evaluated macroscopically for membrane integrity, presence of newly formed bone tissue over and under the membrane, hematoma, integrity of the bone marrow, hole filling, and cortical integrity.

At least one slide from each section, including all experimental time points and treatments, was used for microscopic description. The evaluation allowed for description of the bone defect's characteristics and the interposed tissue's presence and peculiarities, such as granulation tissue, newly formed blood vessels, and areas of bone and/or cartilaginous matrix. Changes in the periosteum, foreign body reaction, and the presence of exogenous material were also assessed.

## Histomorphometry analysis

In the histomorphometric assessment, bone callus area was analyzed at 14 and 30 days. The number of osteoclasts was analyzed in the specimens from 7, 14, and 30 days. The

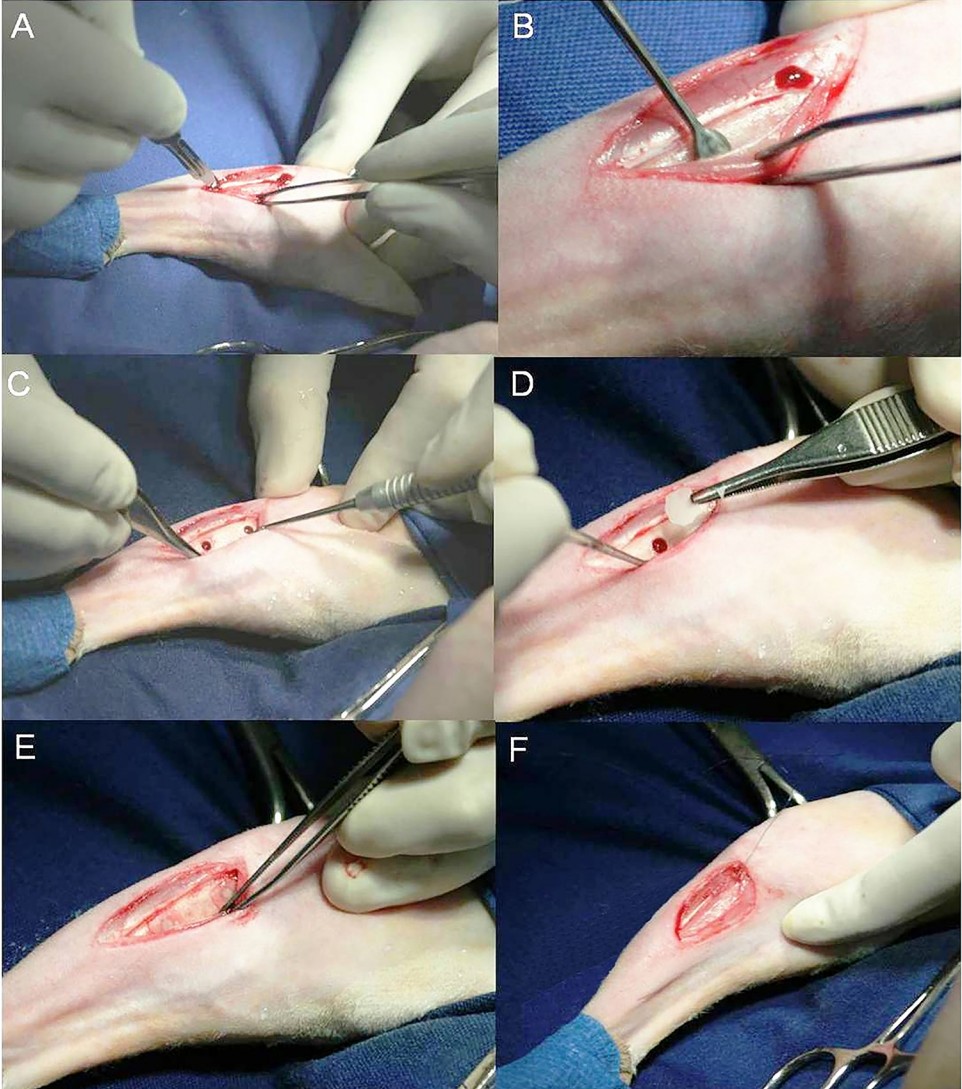

**Fig 1. Surgery for implantation of bioactive polymeric membranes in rabbit tibia defects.** A—Periosteal incision. B—Elevation of periosteal flap with a Molt-type elevator. C—Observe 2.8 mm diameter holes with 10 mm spacing between them. D—Detail of the membrane during placement over the defects. E—Membranes over the holes. F—Suturing of the periosteum over the holes.

quantification of the bone callus involved six sections from each specimen, stained with H&E (Hematoxylin and eosin), randomly selected to measure the bone callus area and cartilaginous matrix area in mm2. In each section, five fields with the bone callus were measured at 40X magnification, referencing to the edge of the bone defect.

The measurements were carried out by a single trained operator using digital morphometry software. The fields were digitized from a conventional light microscope with a CCD camera. The data were entered into a table, and in the end, an average of the fields was obtained, followed by the average of specimens for each group and experimental period.

Osteoclast counting was performed in specimens from the experimental periods of seven, 14, and 30 days. For each section, five fields were measured at 100X magnification, with the edge of the bone defect as the reference point. The slides stained with a tartrate-resistant acid

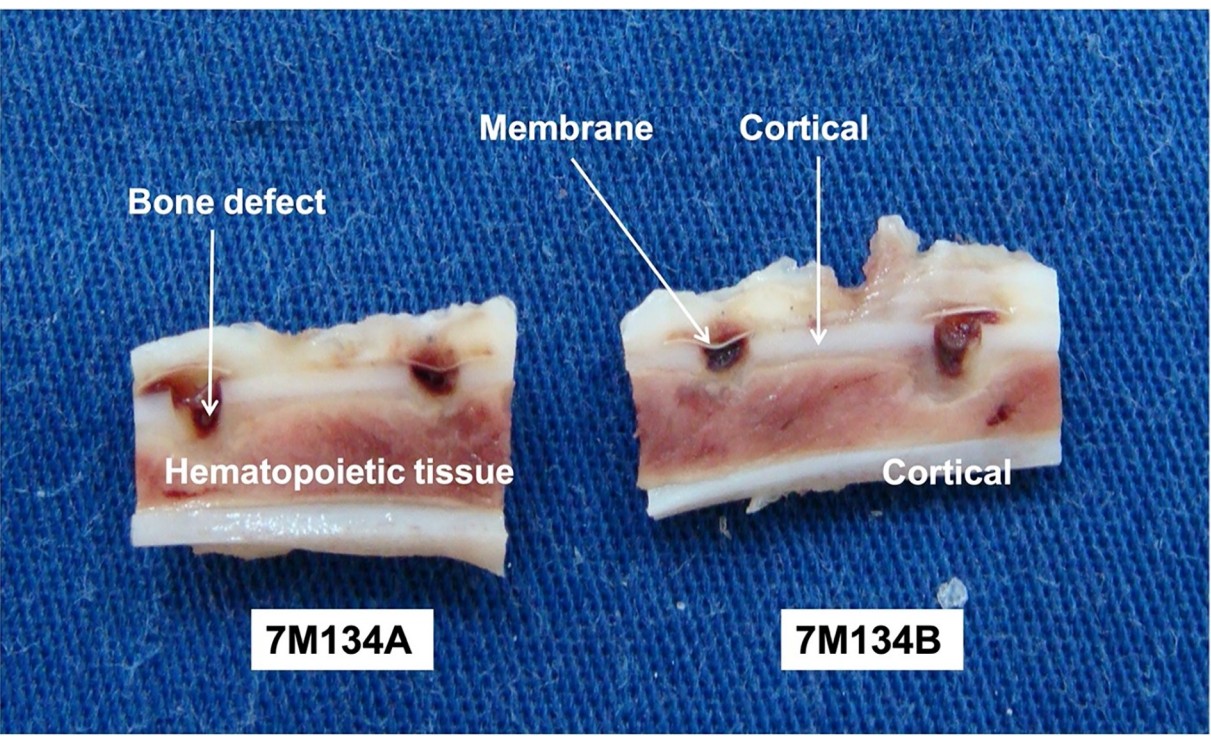

**Fig 2. Section of the specimen along the 'long axis' of the bone.** Note the two sections with two defects each and membranes covering the holes. Below each piece, observe the identification label used throughout the processing.

phosphatase kit (TRAP kit, Sigma Aldrich, St Louis, USA) were used to confirm the presence of osteoclasts during the analysis.

A single trained operator blinded to of the experimental groups of the digital images, performed the measurements using digital morphometry software, with fields digitized by a conventional light microscope equipped with a CCD camera. At the end, the averages of the fields, followed by the average of specimens for each group and experimental period were obtained.

## Statistical analyses

The data on bone callus area and osteoclast numbers were analyzed using the Shapiro-Wilk normality test, which indicated non-parametric and parametric distributions, respectively. To compare the groups at each experimental time point, the Kruskal-Wallis test followed by Dunn's test with Bonferroni adjustments was used for bone callus area. One-way ANOVA followed by Tukey's test was used for osteoclast counting. A significance level of $p < 0.05$ was set for all analyses. Statistical analysis was performed using Bioestat 3.0® software (Federal University of Para, Para, Brazil). After then, the effect sizes were calculated for the experiment using the G*Power Version 3.1.9.7 software [11,12].

## Results

### Histomorphology analysis

Foreign body reaction was absent in all the experimental periods and in the four groups. The histomorphology analysis of specimens from three days revealed a bone defect with homogeneous edges and restricted areas of bone devitalization.

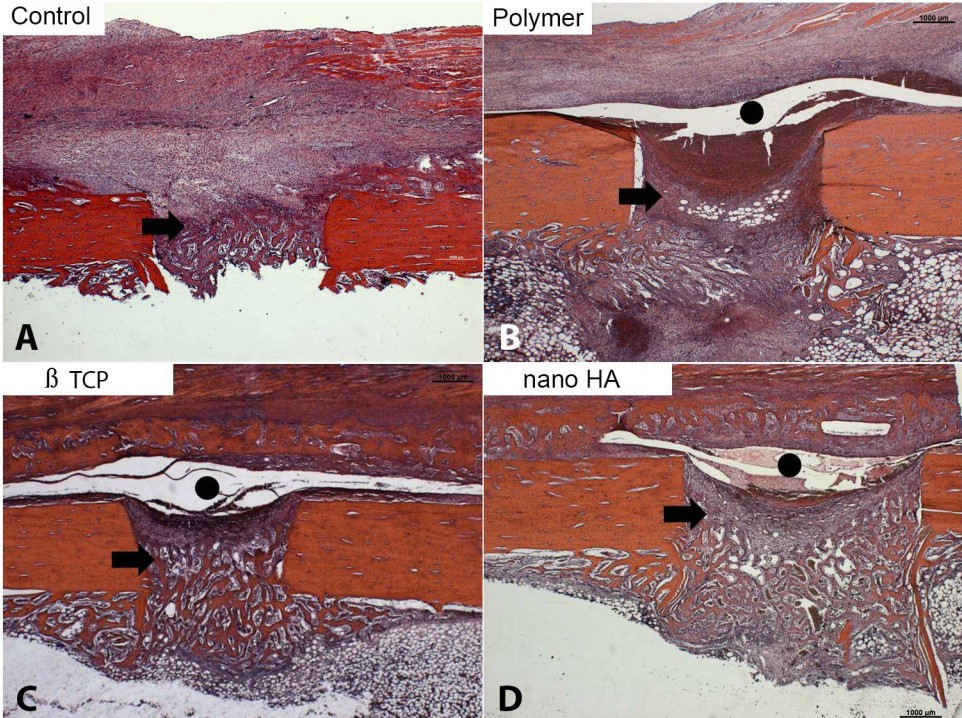

**Fig 3. Initial bone matrix deposition was observed in the defect region in all groups (arrows).** In the membrane groups, this deposition extended beyond the defect. A gap between the periosteum and bone surface was present in these groups, indicating the presence of an exogenous material consistent with the membrane (black circle) (Hematoxylin-eosin, original magnification X 100).

At seven days, the slides from all four groups showed bone defects with homogeneous edges, areas without bone devitalization, periosteal hyperplasia, (particularly in the control group), exuberant granulation tissue with newly formed blood vessels, and focal areas of bone and cartilaginous matrix (Fig 3). In the M1, M2, and M3 groups, the periosteum was separated from the bone matrix due to the presence of the membrane (Fig 3B to 3D).

At 14 days in the control group (Fig 4A), large amount of osteoid material and immature bone trabeculae was observed in the region of the bone defect, as well as hematopoietic tissue in the process of regeneration. In the membrane groups (Fig 4B to 4D), exuberant newly formed bone callus was observed beneath of the bone defect. In th middle of the callus, a space occupied by exogenous material (membrane) was noted. Abundant cartilaginous matrix was present in the callus. The M2 and M3 groups (Fig 4C and 4D) showed clearer endochondral ossification than M1, highlighting the recent replacement of cartilage with bone trabeculae. In the membrane groups, a high frequency of osteoclasts was observed in the bone callus region (Fig 4E).

At 30 days, in the control group, the bone callus showed maturity, consisting mainly of compact bone tissue with advanced remodeling, connecting to the edges of the defect (Fig 5A). In the groups with membranes, the bone callus was exuberant in the defect region, extending over the surface of the preexisting bone. It was composed of cohesive bone trabeculae but with wider medullary spaces than the control (Fig 5B to 5D). The space between the periosteum and the cortex persisted, suggesting the presence of the membranes. Wide medullary spaces filled with hematopoietic material were noted in the defect and preexisting bone surface (Fig 5B to 5D).

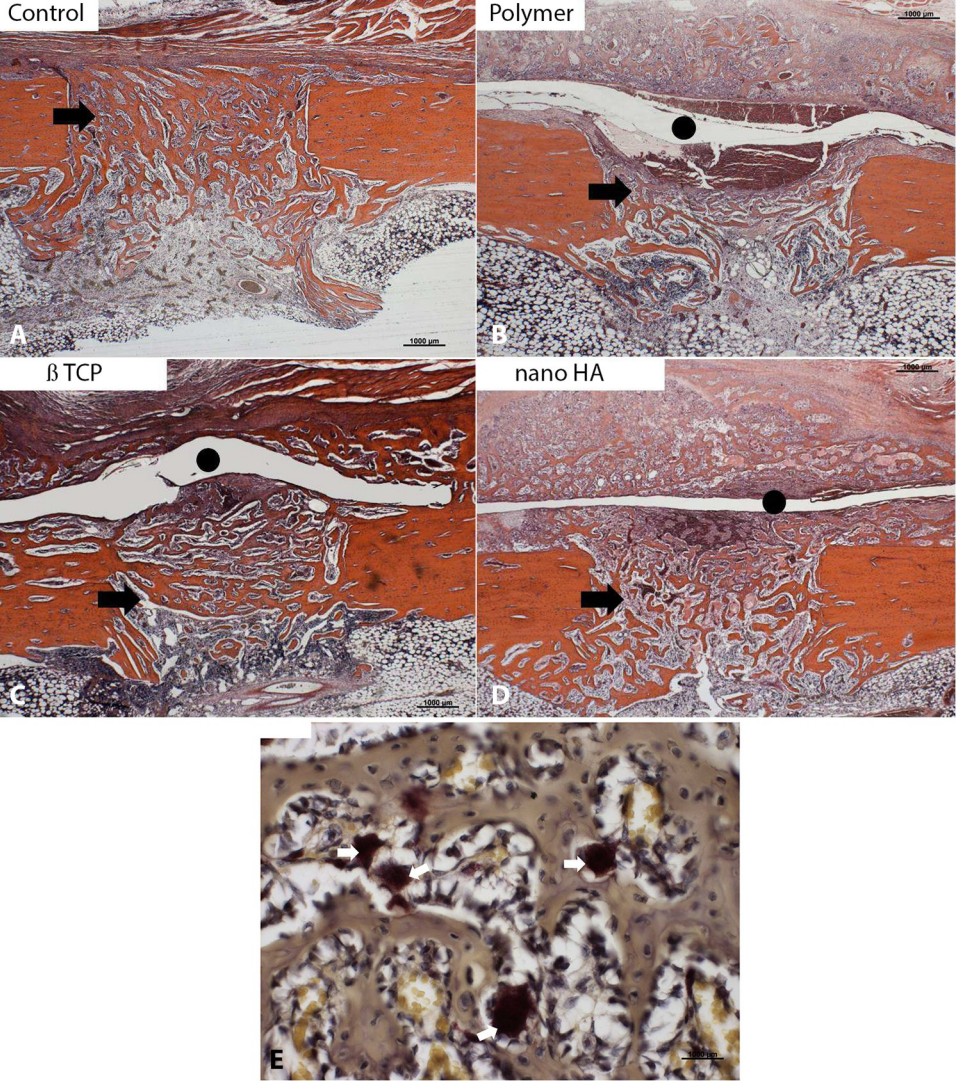

**Fig 4. In the control group, intense bone deposition occurred at the defect site.** A significant bone callus formation (arrows) was observed in the membrane groups, with interconnected trabeculae and robust regeneration of hematopoietic marrow. The membrane space still contained remnants of the exogenous material beneath the periosteum (black circle) (Hematoxylin-eosin, original magnification X 100). Osteoclasts (white arrows) were also observed in the bone callus (E) (TRAP, original magnification 400X).

Analysing the histological findings as a whole, in the control group the bone matrix formation occurred inside the defect. In the membrane groups, there was formation inside the defect and in greater quantity on the surface and periosteum. The presence of the membrane acted as a barrier, isolating the periosteum from the defect, resulting in more bone and cartilaginous matrix formation above the membrane.

## Histomorphometry analysis

The bone callus area was significantly larger in the three membrane groups (M1 –only polymer, M2 –polymer with β-TCP, and M3 –polymer with nano-HA) compared to the control group at both at 14 ($p<0.05$) and 30 days ($p<0.05$) (Fig 6A and 6B). Among the polymers, there were no statistical differences.

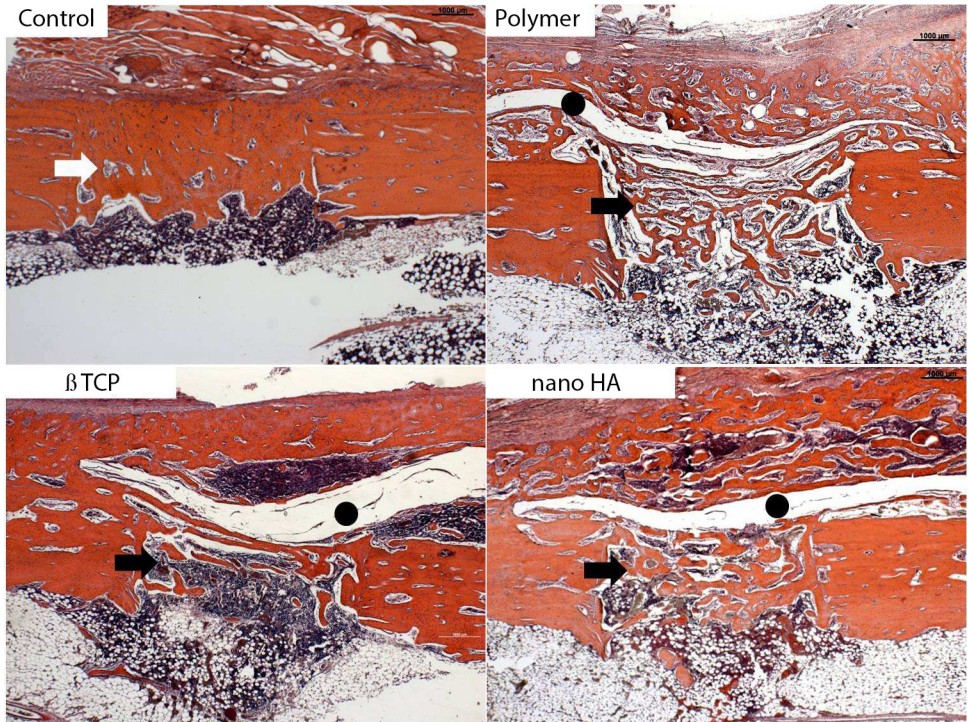

**Fig 5.** Representative histological sections of the bone defect in the control (A) and membrane groups (B to D) 14 days after defect induction. In the control group, the bone callus transformed into cortical bone, indicating complete consolidation. In the membrane groups, the callus showed trabecular bone with hematopoietic marrow (black arrows), with more pronounced remodeling in the nano HA group. The gap containing exogenous material compatible with the membranes (black circle) was still observed in all groups, suggesting incomplete reabsorption of the biomaterial (Hematoxylin-eosin, original magnification X 100).

Regarding to the osteoclast counts, the groups exhibited different trends (Fig 6C). At seven days, the groups M1 and M3 showed a higher number of osteoclasts compared to the control and M2 groups (p<0.01 for all the comparisons). At 14 days, M3 exhibited a higher number of osteoclasts than the other groups (p<0.01 for all the comparisons). At 30 days, the control exhibited a lower number of osteoclasts compared to the membrane groups (p<0.01 for all the comparisons). Among the membrane groups, M3 maintained the highest number of these cells, with significant differences in relation to M1 (p<0.01) (Fig 6A to 6C).

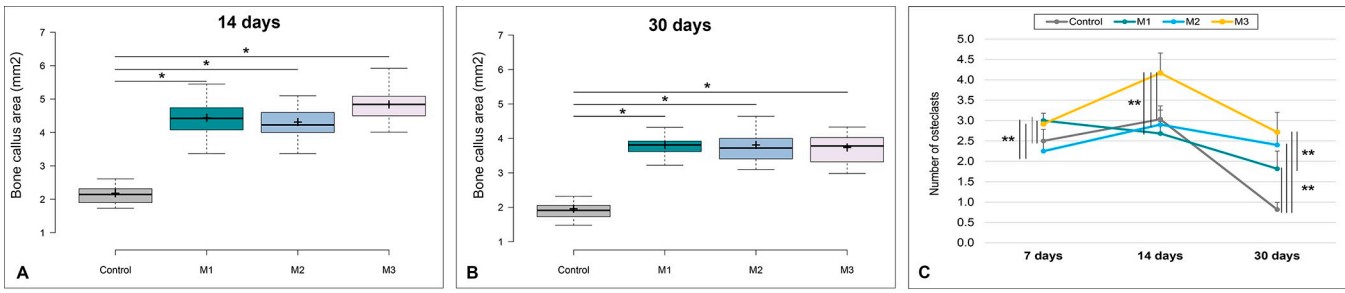

**Fig 6.** Bone callus area for each group at 14 (A) and 30 days (B) after the bone defect induction. Whiskers–represent minimum and maximum values; the horizontal line within the box–median; + mean; box limits– 1$^{st}$ and 3$^{rd}$ percentiles. *p<0.05. p value for Kruskal-Wallis´ test followed by Dunn´s test. C: Mean (± standard deviation) osteoclast counts for each group and experimental times point after bone defect induction. **p<0.01. p values for Anova test, followed by Tukey´s test. M1 –only polymer. M2 –polymer with β-TCP. M3 –polymer with nano-HA. Effect sizes for callus were (f = 0.9508) for 14 days and (f = 0.9479) for 30 days. For osteoclasts, they were (f = 0.8297) for 07 days, (f = 0.8411) for 14 days, and (f = 0.8815) for 30 days.

## Discussion

In the macroscopic evaluation, we observed, at 14 and 30 days, the formation of firm and dense tissue between the membrane and the periosteum, which was absent in the controls. This suggests greater bone formation near the biomaterial, indicating the osteoinductive and osteoconductive potential of the bioactive polymers [13–16].

Throughout the four experimental periods, we did not observe sustained inflammatory reactions, signs of toxicity, or mutagenic evidence, indicating the remarkable biocompatibility of the polymeric biomaterials [17–19]. These characteristics, crucial for recommending biomaterials in clinical situations, align with the desirable properties mentioned in the literature, which include biocompatibility, bioactivity, predictability, ease of clinical application, and absence of toxicity [17,20]. In contrast, castor oil polymer, although extensively studied [21,22], raised concerns about moderate to intense inflammatory reactions, as well as toxic effects of ricin and ricinine [23–25].

At seven days, all slides exhibited abundant granulation tissue, especially in the polymer groups, accompanied by a significant proliferation of newly formed blood vessels. Vascularization is essential for the formation of mineralized bone matrix, ensuring adequate oxygen tension at the fracture site, crucial for the differentiation of osteoblasts and the overall repair process [3,26,27].

Observing the work of PENG *et al.* (2005) [26], FERRIGNO, DELLA NINA & FANTONI (2007) [28] and ROZEN et al. 2007 [3], the results can be extrapolated to routine orthopedic situations in dogs and cats, especially in non-unions. Non-unions are common in small dogs, in the distal region of the tibia and radius and ulna, where muscular coverage and vascularization are limited [29]. The observation of moderate to intense angiogenesis in the polymer groups at seven and 14 days is encouraging for the treatment of non-unions with vascular compromise [25].

At seven and 14 days, in the control group, focal areas of bone and cartilaginous matrix with osteoclasts in the osteoid matrix region were observed, indicating that the repair process was occurring as expected in untreated situations [1,3,4,30]. Compared to the control, specimens that received polymers exhibited a greater amount of bone and cartilaginous matrix with osteoclasts. In this period, the biomaterials showed osteoinductive potential, stimulating remodeling with a significant observation of osteoclasts. Among the polymers, the formation of cartilaginous matrix was more pronounced in those that received polymer and nano-HA. These results may be related to the bioactive characteristic of the polymers, as highlighted in the literature [14,17,31,32].

Nano-HA has demonstrated osteoinductive properties in studies, creating a healthy environment for bone tissue regeneration [25,33]. Both β-tricalcium phosphate and nano-HA promote bone formation around their surfaces. However, nano-HA holds an advantage in terms of osteoinductivity, meaning it can induce the differentiation of stem cells into osteoblasts more effectively [25,34].

At 14 days, all groups showed an abundance of osteoclasts, decreasing at 30 days, indicating more intense remodeling in the initial period. At 30 days, the control group had essentially completed the process, while the groups with polymeric membranes maintained moderate activity. Regardless of the group, the repair followed a similar pattern to what occurs naturally, as described in other previously published studies [3,4,6,35].

The peculiarity of this experiment, considering typiccally reactive individuals, may explain the normal repair in the control group animals. However, those who received biomaterials showed excessive bone callus formation and slower remodeling. BOYLE, SIMONET & LACEY (2003) [36] e SCHINDELER et al. (2008) [4] indicate that remodeling persists as long

as there is bone tissue deposition. Assuming osteoinductive effects in hypo-reactive individuals is relevant, but studies in patients with slow bone repair are essential to support clinical use.

At 14 days, endochondral ossification was more evident in the polymer groups, especially those with β-TCP and nano-HA. Despite the significant presence of cartilaginous matrix, recent cartilage replacement by bone trabeculae was observed. The polymers, especially those with β-TCP or nano-HA, appear to promote bone neoformation and accelerate the repair process [28,34,37,38], which has also been confirmed in more recent studies [25,33].

Based on the literature consulted it is known that the differentiation of mesenchymal cells into osteoblasts is accompanied by the synthesis of the extracellular matrix and bone proteins, leading to the mineralization of the matrix [9]. The TGF-β1 is involved in every phase of the osteogenic process, and its tissue levels are directly proportional to bone formation [3,39]. Proliferation and differentiation of fibroblasts and chondrocytes are coordinated by growth factors, including TGF-β2 and β3, FGF-1, and IGF. Recent studies have also indicated that other factors, such as BMP-2 and BMP-7, play critical roles in enhancing osteogenic differentiation and are being explored for their therapeutic potential in bone regeneration [40]. In response to these factors, chondrocytes produce extracellular matrix proteins, especially type II collagen or type X collagen for chondrocyte hypertrophy [2–4,30]. The study did not aim to elucidate the osteoinductive mechanisms of the polymers, but based on the literature, it is reasonable to assume that osteoinduction is linked to the mentioned events.

At 30 days, in the polymer groups, we observed bone callus in full remodeling stage, indicated by the presence of active osteoclasts in Howship's lacunae. This scenario suggests a significant amount of cancellous bone, maintaining the remodeling process due to increased bone deposition [5,41,42]. This fact suggests probable complete remodeling over a period longer than that observed in the experiment. However, the maximum observation period was 30 days, emphasizing the need for studies with a longer observation period to monitor the remodeling and absorption of the membrane. In the control group, at 30 days, the bone callus was practically remodeled.

At seven, 14, and 30 days, in the control group, bone matrix formed inside the defect, and in the membrane groups, both inside the defect and in large quantities on the surface of the hole and on the periosteum. The presence of the membrane acted as a barrier that isolated the periosteum from the bone defect, resulting in a large amount of bone and cartilaginous matrix formation above the membrane, unlike the control group, where this response was more homogeneous and localized. These results indicate the stimulation of bone formation by the polymers with or without the addition of β-TCP or nano-HA. The osteoinductive/osteoconductive potential of these polymers has been discussed by several referenced authors [29,32,33,38,43–45], with bone formation through osteoinduction and osteoconduction being dependent on the recruitment of mesenchymal precursors that undergo various proliferative phases before differentiation into a phenotype that supports matrix deposition and mineralization [14,32].

In the context of bone callus formation restricted to the defect, the control group exhibited a more robust bone formation with cohesive and homogeneous trabeculae compared to the polymer groups. As discussed earlier, a powdered or gel-like biomaterial may facilitate osteoinduction within the defect, avoiding the isolation of the periosteum, a rich source of osteoprogenitor cells [3,4].

The absorption of the membrane was not completed after 30 days of observation, indicating possible mechanical fragility in the space occupied by the membrane. Additionally, the bone callus was not completely remodeled. Although a remodeled callus has superior mechanical properties, a larger callus can provide similar mechanical performance [4,7].

The polymeric membranes, especially those with nano-HA, demonstrated greater bone formation with adequate stimulation for remodeling in our study. Proportional increases in the number of osteoclasts and the percentage of bone trabeculae with osteoclasts were observed in these cases. No foreign body reactions, signs of toxicity, or mutagenic activities were evident in the evaluated periods. Thus, based on the work of SZABO et al. (2005) [17], RORIZ et al. (2006) [18] e NANDI et al. (2010) [19], we can recommend their use in clinical situations where attenuated bone repair is observed or expected, such as non-unions, bone defects, osteotomies, arthrodesis, and metabolic deficiencies resulting in delayed bone consolidation [14].

The overall evaluation of the results highlights the need for further studies to assess the clinical applicability of the polymers. Investigating other physical presentations, complete absorption of the polymers, and mechanical tests may provide valuable insights into the science of biomaterials.

## Conclusion

All three membranes exhibited significant chondro and osteoinductive properties, with the membrane containing nano-hydroxyapatite showing the highest level of osteoinductive potential. While the membranes were not fully absorbed, there was evident intense remodeling activity. Additional research with extended experimental periods is needed to assess the mechanical characteristics of the newly formed bone and the membrane absorption process.

## Supporting information

**S1 Table. Raw data for bone callus measurements.** The bone callus area (in mm$^2$) was quantified using at least six sections per specimen from each treatment group at the 14- and 30-day time points.
(DOCX)

**S2 Table. Raw data for osteoclast counts.** Mean and standard deviation (SD) of the number of osteoclasts for each treatment at the 7-, 14- and 30-day time points.
(DOCX)

## Author Contributions

**Conceptualization:** Olicies da Cunha, Cássio Ricardo Auada Ferrigno, Solimar Dutra da Silveira, Bruno Gregnanin Pedron, Danielle Tiemi Komorizono, Flávia Cristina Rosin Prado, Walter Israel Rojas Cabrera, Luciana Corrêa.

**Data curation:** Olicies da Cunha, Cássio Ricardo Auada Ferrigno, Solimar Dutra da Silveira, Bruno Gregnanin Pedron, Danielle Tiemi Komorizono, Flávia Cristina Rosin Prado, Walter Israel Rojas Cabrera, Luciana Corrêa.

**Formal analysis:** Olicies da Cunha, Cássio Ricardo Auada Ferrigno, Solimar Dutra da Silveira, Bruno Gregnanin Pedron, Danielle Tiemi Komorizono, Flávia Cristina Rosin Prado, Walter Israel Rojas Cabrera, Luciana Corrêa.

**Funding acquisition:** Olicies da Cunha, Cássio Ricardo Auada Ferrigno, Solimar Dutra da Silveira, Bruno Gregnanin Pedron, Danielle Tiemi Komorizono, Flávia Cristina Rosin Prado, Walter Israel Rojas Cabrera, Luciana Corrêa.

**Investigation:** Olicies da Cunha, Cássio Ricardo Auada Ferrigno, Solimar Dutra da Silveira, Bruno Gregnanin Pedron, Danielle Tiemi Komorizono, Flávia Cristina Rosin Prado, Walter Israel Rojas Cabrera, Luciana Corrêa.

**Methodology:** Olicies da Cunha, Cássio Ricardo Auada Ferrigno, Solimar Dutra da Silveira, Bruno Gregnanin Pedron, Danielle Tiemi Komorizono, Flávia Cristina Rosin Prado, Walter Israel Rojas Cabrera, Luciana Corrêa.

**Project administration:** Olicies da Cunha, Cássio Ricardo Auada Ferrigno, Solimar Dutra da Silveira, Bruno Gregnanin Pedron, Danielle Tiemi Komorizono, Flávia Cristina Rosin Prado, Walter Israel Rojas Cabrera, Luciana Corrêa.

**Resources:** Olicies da Cunha, Cássio Ricardo Auada Ferrigno, Solimar Dutra da Silveira, Bruno Gregnanin Pedron, Danielle Tiemi Komorizono, Flávia Cristina Rosin Prado, Walter Israel Rojas Cabrera, Luciana Corrêa.

**Software:** Olicies da Cunha, Cássio Ricardo Auada Ferrigno, Solimar Dutra da Silveira, Bruno Gregnanin Pedron, Danielle Tiemi Komorizono, Flávia Cristina Rosin Prado, Walter Israel Rojas Cabrera, Luciana Corrêa.

**Supervision:** Olicies da Cunha, Cássio Ricardo Auada Ferrigno, Solimar Dutra da Silveira, Bruno Gregnanin Pedron, Danielle Tiemi Komorizono, Flávia Cristina Rosin Prado, Walter Israel Rojas Cabrera, Luciana Corrêa.

**Validation:** Olicies da Cunha, Cássio Ricardo Auada Ferrigno, Solimar Dutra da Silveira, Bruno Gregnanin Pedron, Danielle Tiemi Komorizono, Flávia Cristina Rosin Prado, Walter Israel Rojas Cabrera, Luciana Corrêa.

**Visualization:** Olicies da Cunha, Cássio Ricardo Auada Ferrigno, Solimar Dutra da Silveira, Bruno Gregnanin Pedron, Danielle Tiemi Komorizono, Flávia Cristina Rosin Prado, Walter Israel Rojas Cabrera, Luciana Corrêa.

**Writing – original draft:** Olicies da Cunha, Cássio Ricardo Auada Ferrigno, Solimar Dutra da Silveira, Bruno Gregnanin Pedron, Danielle Tiemi Komorizono, Flávia Cristina Rosin Prado, Walter Israel Rojas Cabrera, Luciana Corrêa.

**Writing – review & editing:** Olicies da Cunha, Cássio Ricardo Auada Ferrigno, Solimar Dutra da Silveira, Bruno Gregnanin Pedron, Danielle Tiemi Komorizono, Flávia Cristina Rosin Prado, Walter Israel Rojas Cabrera, Luciana Corrêa.

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
