## [Editor Report · Decision Letter 0]

20 Aug 2024

PONE-D-24-33424Biological interaction of bioactive polymeric membranes in induced bone defects in rabbit tibiasPLOS ONE

Dear Dr. Ferrigno,

Thank you for submitting your manuscript to PLOS ONE. After careful consideration, we feel that it has merit but does not fully meet PLOS ONE’s publication criteria as it currently stands. Therefore, we invite you to submit a revised version of the manuscript that addresses the points raised during the review process.

This article presents several flaws that have been highlighted in the report below point by point.The pitfalls are related either to the Methods and to some specific assays (time points, controls, comparisons of data), as well as to other parts including Discussion and References that are too old and not up to dated.The Authors must answer to each point raised and send the revised version along with the point to point rebuttal letter answers and in red the manuscript amendments.==============================

We look forward to receiving your revised manuscript.

Kind regards,

Gianpaolo Papaccio, M.D., Ph.D.

Academic Editor

PLOS ONE

Journal Requirements:

2. To comply with PLOS ONE submissions requirements, in your Methods section, please provide additional information regarding the experiments involving animals and ensure you have included details on (1) methods of sacrifice, (2) methods of anesthesia and/or analgesia, and (3) efforts to alleviate suffering

"The research was paartially funded by FUndacao de Amparo a Pesquisa do Estado de São Paulo.

The funding pay for the animals used in the project, and the disposeble use in the project"

Additional Editor Comments:

This study intends to assess bone repair using three osteoinductive polymers in bone defects induced in rabbit tibias.

It is not new for the topic.

A general interest can be found for the specific research but the study and manuscript present several flaws.

The article mentions using 48 rabbits, but there is no clear justification for this sample size. A power analysis or rationale for selecting this number of subjects would be essential to ensure the study is adequately powered to detect differences.

The control group is described as "without biomaterial," but it lacks detailed information on how the control condition was maintained. For instance, were the defects in the control group left untreated, or were they treated with a standard material? Clarification is needed here.

The choice of time points (3, 7, 14, and 30 days) should be justified in the context of bone healing stages. The article does not adequately explain why these particular time points were selected or how they relate to the expected timeline of bone regeneration.

The article lacks a detailed explanation of the statistical tests used. For instance, it’s not clear if the data distribution was assessed before applying parametric or non-parametric tests. The choice of tests should be justified, and any assumptions made should be stated clearly.

The study compares several groups at different time points. It is unclear if adjustments were made for multiple comparisons to control for the risk of Type I errors (false positives).

The results section states that the bone callus area was significantly smaller in the control group compared to the membrane groups, and that M3 had a greater bone formation than M2 at 14 days. However, there is little discussion on the biological mechanisms that might explain these observations. A deeper exploration of why nano-hydroxyapatite outperformed β-tricalcium phosphate would strengthen the manuscript.

Terms like "significantly smaller" and "greater bone formation" are used without providing the exact statistical values (e.g., p-values, confidence intervals). Including these details would add precision to the claims.

The conclusions drawn seem somewhat overgeneralized based on the data presented. For example, claiming that the osteoinductive potential of nano-hydroxyapatite is superior to β-tricalcium phosphate should be tempered with acknowledgment of the study’s limitations and the need for further research.

The discussion section does not adequately address the limitations of the study. Potential biases, such as the lack of blinding in the histomorphometric analysis or any variations in the surgical procedures, should be acknowledged.

While the manuscript mentions ethical approval by the Ethics Committee on Animal Use, the description is brief. It would be beneficial to provide more details on the ethical considerations taken, particularly concerning animal welfare and how it was ensured throughout the study.

The figures and graphs should be clear and adequately labeled. It’s essential to ensure that all axes are labeled with units, and that error bars are explained in the figure legends.

Although the manuscript states it has undergone professional proofreading, there are still minor grammatical errors and awkward phrasing that can hinder readability. A final review by a native English speaker or professional editor might improve the overall presentation.

Some references, like the one from Martin, Gooi, & Sims (2009), are relatively outdated for a rapidly evolving field like biomedical engineering. Including more recent literature would make the manuscript more robust.

The importance of biocompatibility as well as the role exerted by some cytotypes like MSCs and new biomaterials are lacking: see and discuss (2015) Dental Materials, 31 (3) , pp. 235-243 as well as papers like (2024) Advanced Functional Materials 34(30),2400766 for new materials.

Additionally, there are only a few references cited in key sections, which may give the impression that the discussion is not fully supported by the literature.

Other references are too old and only the more recent must be used.

---

## [Author Response · Author response to Decision Letter 0]

18 Oct 2024

Dear Academic Editor and Reviewers,

We would like to express our gratitude for your attention to our manuscript.

Below, you will find our responses to each of your requests.

The changes made to the manuscript in response to these requests are highlighted in red.

Yours sincerely,

The Authors

RESPONSES TO REVIEWERS

1 - Please ensure that your manuscript meets PLOS ONE's style requirements, including those for file naming. The PLOS ONE style templates can be found at…

R: We have reviewed the manuscript and made all necessary adjustments to ensure it complies with PLOS ONE's style requirements, including proper file naming. We have used the PLOS ONE style templates provided in the link mentioned.

2 -To comply with PLOS ONE submissions requirements, in your Methods section, please provide additional information regarding the experiments involving animals and ensure you have included details on (1) methods of sacrifice, (2) methods of anesthesia and/or analgesia, and (3) efforts to alleviate suffering.

R: Thank you for raising these points. Below are the requested revisions to the Methods section, in accordance with PLOS ONE submission requirements.

(1) – Ln 117– “…histological analyses. At each evaluation time point, the animals were sedated with intramuscular ketamine (10 mg/kg) and midazolam (1 mg/kg). Anesthesia induction was performed with sodium thiopental (50 mg/kg) intravenously, followed by euthanasia with a lethal dose of potassium chloride using the same route.”

(2) – Ln 100 – “The animals were premedicated intramuscularly with ketamine (30 mg/kg) and midazolam (1 mg/kg), followed by mask induction with isoflurane. All animals underwent endotracheal intubation, and anesthesia was maintained with isoflurane in oxygen in a non-rebreathing circuit to achieve a surgical plane of anesthesia characterized by muscle relaxation, ventral ocular rotation and absence of palpebral reflex. Sacrococcygeal epidural lidocaine 2% (0.3ml/kg) provided surgical analgesia.

(3) – Ln 78 – “The animals underwent an adaptation period in a ventilated environment in individual cages, with free access to water and commercial feed. The environment was sanitized daily.”

Ln 114 – “At the end of the surgical procedure, all animals received ketoprofen (3mg/kg) and morphine (2mg/kg) subcutaneously for 5 and 3 days, respectively.”

3 - We note that the grant information you provided in the ‘Funding Information’ and ‘Financial Disclosure’ sections do not match.

R: We apologize for this mistake. It has been corrected.

4 - Thank you for stating the following financial disclosure:

R: We thank you for suggesting. It has been corrected.

"The funders had no role in study design, data collection and analysis, decision to publish, or manuscript preparation."

5 - When completing the data availability statement of the submission form, you indicated that you will make your data available on acceptance. We strongly recommend all authors decide on a data sharing plan before acceptance, as the process can be lengthy and hold up publication timelines. Please note that, though access restrictions are acceptable now, your entire data will need to be made freely accessible if your manuscript is accepted for publication. This policy applies to all data except where public deposition would breach compliance with the protocol approved by your research ethics board. If you are unable to adhere to our open data policy, please kindly revise your statement to explain your reasoning and we will seek the editor's input on an exemption. Please be assured that, once you have provided your new statement, the assessment of your exemption will not hold up the peer review process.

R: We thank you for suggesting. All data was made available, but there was an error in filling out the form. The mistake has already been corrected.

ADDITIONAL EDITOR COMMENTS 

The article mentions using 48 rabbits, but there is no clear justification for this sample size. A power analysis or rationale for selecting this number of subjects would be essential to ensure the study is adequately powered to detect differences.

R: Thank you for pointing out this detail. A test to determine sample size was not performed during experiment planning, but a power analysis of the data obtained was performed. We inserted the following sentences:

Ln 170 – “After then, effect sizes were calculated for the experiment using the G*Power Version 3.1.9.7 software [11, 12].”

Ln 378 - “Effect sizes for callus were (f = 0.9508) for 14 days and (f = 0.9479) for 30 days. For osteoclasts, they were (f = 0.8297) for 07 days, (f = 0.8411) for 14 days, and (f = 0.8815) for 30 days.”

References: 

Ln 429 – “11. Faul F, Erdfelder E, Buchner A, Lang AG. Statistical power analyses using G*Power 3.1: Tests for correlation and regression analyses. Behavior Research Methods. 2009; 41:1149-1160.”

Ln 432 – “12. Cohen J. Statistical power analysis for the behavioral sciences. 2nd ed. Hillsdale: Lawrence Erlbaum Associates, 1988:1-567.”

The control group is described as "without biomaterial," but it lacks detailed information on how the control condition was maintained. For instance, were the defects in the control group left untreated, or were they treated with a standard material? Clarification is needed here.

R: We appreciate your relevant observation regarding the description of the control group in our study. To clarify, no type of material was added to the defect in the control group. We inserted the following sentences:

Ln 95 – “The animals in the control group were subjected to the same anesthetic, pre-operative, transoperative, and post-surgical protocols. The only difference was the absence of any material in the created defects.”

The choice of time points (3, 7, 14, and 30 days) should be justified in the context of bone healing stages. The article does not adequately explain why these particular time points were selected or how they relate to the expected timeline of bone regeneration.

R: Thank you for your question. These experimental times were assigned considering the three main bone regeneration phases (inflammatory, productive, and remodelling phases). We inserted the following sentences in the Methods item (2nd paragraph): 

Ln 82 - “The experimental times of three, seven, 14, and 30 days were designated to assess the potential inflammatory reaction induced by the biomaterials, particularly the foreign body reaction. The 14-day period was specifically chosen for morphometric analysis to evaluate the osteoinduction potential of the material, as this stage is critical in the bone regeneration process due to increased cell proliferation. The 30-day time frame was allocated for assessing the initial phase of bone remodeling induced by the membranes and to determine whether there was resorption of the biomaterial by osteoclasts or other types of phagocytic cell.”

We inserted figures showing the histological aspect as a complement to the results.

The article lacks a detailed explanation of the statistical tests used. For instance, it’s not clear if the data distribution was assessed before applying parametric or non-parametric tests. The choice of tests should be justified, and any assumptions made should be stated clearly. The study compares several groups at different time points. It is unclear if adjustments were made for multiple comparisons to control for the risk of Type I errors (false positives).

R: Type I errors were controlled by multiple comparisons using the Kruskal-Wallis test and the Anova test. Additional details of the statistical tests can be found in the last paragraph of the Methods section:

Ln 164 - “The data on bone callus area and osteoclast numbers were analyzed using the Shapiro-Wilk normality test, which indicated non-parametric and parametric distributions, respectively. To compare the groups at each experimental time point, the Kruskal-Wallis test followed by Dunn's test with Bonferroni adjustments was used for bone callus area. One-way ANOVA followed by Tukey's test was used for osteoclast counting. A significance level of p<0.05 was set for all analyses. Statistical analysis was performed using Bioestat 3.0® software (Federal University of Para, Para, Brazil).”

To clarify the results, we also inserted a graph showing the bone callus and osteoclast count statistics (please see Fig. 6).

The results section states that the bone callus area was significantly smaller in the control group compared to the membrane groups, and that M3 had a greater bone formation than M2 at 14 days. However, there is little discussion on the biological mechanisms that might explain these observations. A deeper exploration of why nano-hydroxyapatite outperformed β-tricalcium phosphate would strengthen the manuscript.

R: Thank you for this valuable observation. We have incorporated some new references (16, 25, 33, 34 e 40) and added the following sentences in the Discussion.

Ln 253 – “Nano-HA has demonstrated osteoinductive properties in studies, creating a healthy environment for bone tissue regeneration [25, 33]. Both β-tricalcium phosphate and nano-HA promote bone formation around their surfaces. However, nano-HA holds an advantage in terms of osteoinductivity, meaning it can induce the differentiation of stem cells into osteoblasts more effectively regeneration [25, 33].”

Ln 269 – “At 14 days, endochondral ossification was more evident in the polymer groups, especially those with β-TCP and nano-HA. Despite the significant presence of cartilaginous matrix, recent replacement of cartilage by bone trabeculae was observed. The polymers, especially those with β-TCP or nano-HA, appear to promote bone neoformation and accelerate the repair process [28, 34, 37], which has also been confirmed in more recent studies [25, 33].”

Ln 279 – “Recent studies have also indicated that other factors, such as BMP-2 and BMP-7, play critical roles in enhancing osteogenic differentiation and are being explored for their therapeutic potential in bone regeneration [40].”

Terms like "significantly smaller" and "greater bone formation" are used without providing the exact statistical values (e.g., p-values, confidence intervals). Including these details would add precision to the claims.

R: We inserted the p values attached to these expressions. Please check in the manuscript the corrections highlighted in red, mainly in Results section.

The conclusions drawn seem somewhat overgeneralized based on the data presented. For example, claiming that the osteoinductive potential of nano-hydroxyapatite is superior to β-tricalcium phosphate should be tempered with acknowledgment of the study’s limitations and the need for further research.

R: We rephrase the Conclusions: 

Ln 331 - “All three membranes exhibited significant chondro and osteoinductive properties, with the membrane containing nano-hydroxyapatite showing the highest level of potential. While the membranes were not fully absorbed, there was evident intense remodeling activity. Additional research with extended experimental periods is needed to assess the mechanical characteristics of the newly formed bone and the membranes absorption process.”

The discussion section does not adequately address the limitations of the study. Potential biases, such as the lack of blinding in the histomorphometric analysis or any variations in the surgical procedures, should be acknowledged.

R: Thank you for this important observation. The morphometry analysis was performed without the knowledge of the experimental groups. We inserted this observation in the Material and Methods, “Histomorphometry analysis” section (last paragraph): 

Ln 157 – “A single trained operator, blinded to of the experimental groups of the digital images,…”

We inserted other limitations in the Discussion section (last paragraph): “The overall evaluation of the results highlights the need for further studies to assess the clinical applicability of the polymers. Investigating other physical presentations, complete absorption of the polymers, and mechanical tests may provide valuable insights into the science of biomaterials.”

While the manuscript mentions ethical approval by the Ethics Committee on Animal Use, the description is brief. It would be beneficial to provide more details on the ethical considerations taken, particularly concerning animal welfare and how it was ensured throughout the study.

R: Thank you for pointing out this detail.

We inserted this information on Study design.

Ln 78 – “The animals underwent an adaptation period in a ventilated environment in individual cages, with free access to water and commercial feed. The environment was sanitized daily.”

And

LN 114 – “At the end of the surgical procedure, all animals received ketoprofen (3mg/kg) and morphine (2mg/kg) subcutaneously for 5 and 3 days, respectively.”

The figures and graphs should be clear and adequately labeled. It’s essential to ensure that all axes are labeled with units, and that error bars are explained in the figure legends.

R: Thank you for this important observation. To improve the clarity of our results, we have added new figures (3, 4 and 5) and the graph was changed (Figure 6). All illustrations contain detailed legends with comprehensive explanations.

Ln 338 – “Captions of the figures”

“Fig. 3. Initial bone matrix deposition was observed in the defect region in all groups (arrows). In the membrane groups, this deposition extended beyond the defect. A gap between the periosteum and bone surface was present in these groups, indicating the presence of an exogenous material consistent with the membrane (black circle) (Hematoxylin-eosin, original magnification X 100).”

“Fig. 4. In the control group, intense bone deposition occurred at the defect site. A significant bone callus formation (arrows) was observed (arrows) in the membrane groups, with interconnected trabeculae and robust regeneration of hematopoietic marrow. The membrane space still contained remnants of the exogenous material beneath the periosteum (black circle) (Hematoxylin-eosin, original magnification X 100). Osteoclasts (white arrows) were also observed in the bone callus (E) (TRAP, original magnification 400X).”

“Fig. 5. Representative histological sections of the bone defect in the control (A) and membrane groups (B to D) after 14 days after defect induction. In the control group, the bone callus transformed into cortical bone, indicating complete consolidation. In the membrane groups, the callus showed trabecular bone with hematopoietic marrow (black arrows), with more pronounced remodeling in the nano HA group. The gap containing exogenous material compatible with the membranes (black circle) was still observed in all groups, suggesting incomplete reabsorption of the biomaterial (Hematoxylin-eosin, original magnification X 100).”

“Fig. 6. Bone callus area for each group at 14 (A) and 30 days (B) after the bone defect induction. Whiskers – represent minimum and maximum values; the horizontal line within the box – median; + mean; box limits – 1st and 3rd percentiles. �p<0.05. p value for Kruskal-Wallis´ test followed by Dunn´s test. C: Mean (� standard deviation) osteoclast counts for each group and experimental times point after bone defect induction. ��p<0.01. p values for Anova test, followed by Tukey´s test. M1 – only polymer. M2 – polymer with β-TCP. M3 – polymer with nano-HA. 

Effect sizes for callus were (f = 0.9508) for 14 days and (f = 0.9479) for 30 days. For osteoclasts, they were (f = 0.8297) for 07 days, (f = 0.8411) for 14 days, and (f = 0.8815) for 30 days.”

Although the manuscript states it has undergone professional proofreading, there are still minor grammatical errors and awkward phrasing that can hinder readability. A final review by a

---

## [Decision Letter · Decision Letter 1]

1 Nov 2024

Biological interaction of bioactive polymeric membranes in induced bone defects in rabbit tibias

PONE-D-24-33424R1

Dear Dr. Ferrigno,

We’re pleased to inform you that your manuscript has been judged scientifically suitable for publication and will be formally accepted for publication once it meets all outstanding technical requirements.

Kind regards,

Gianpaolo Papaccio, M.D., Ph.D.

Academic Editor

PLOS ONE

Additional Editor Comments (optional):

Reviewers' comments:

Reviewer's Responses to Questions

**Comments to the Author**

1. If the authors have adequately addressed your comments raised in a previous round of review and you feel that this manuscript is now acceptable for publication, you may indicate that here to bypass the “Comments to the Author” section, enter your conflict of interest statement in the “Confidential to Editor” section, and submit your "Accept" recommendation.

Reviewer #1: All comments have been addressed

2. Is the manuscript technically sound, and do the data support the conclusions?

Reviewer #1: Yes

3. Has the statistical analysis been performed appropriately and rigorously? 

Reviewer #1: Yes

4. Have the authors made all data underlying the findings in their manuscript fully available?

Reviewer #1: Yes

5. Is the manuscript presented in an intelligible fashion and written in standard English?

Reviewer #1: Yes

6. Review Comments to the Author

Reviewer #1: The Authors have fully answered point by point all previous concerns. The Authors have clarified and modified the manuscript in each point as requested by reviewers.

7. PLOS authors have the option to publish the peer review history of their article (what does this mean?). If published, this will include your full peer review and any attached files.

Reviewer #1: No

---

## [Editor Report · Acceptance letter]

25 Nov 2024

PONE-D-24-33424R1 

PLOS ONE

Dear Dr. Ferrigno, 

I'm pleased to inform you that your manuscript has been deemed suitable for publication in PLOS ONE. Congratulations! Your manuscript is now being handed over to our production team.

Kind regards, 

on behalf of

Prof. Gianpaolo Papaccio 

Academic Editor

PLOS ONE